# Rare Yeasts in Latin America: Uncommon Yet Meaningful

**DOI:** 10.3390/jof9070747

**Published:** 2023-07-14

**Authors:** Óscar Gil, Juan Camilo Hernández-Pabón, Bryan Tabares, Carlos Lugo-Sánchez, Carolina Firacative

**Affiliations:** 1Group MICROS Research Incubator, School of Medicine and Health Sciences, Universidad de Rosario, Bogota 111221, Colombia; oscari.gil@urosario.edu.co (Ó.G.); juancamilo.hernand04@urosario.edu.co (J.C.H.-P.); bryan.tabares@urosario.edu.co (B.T.); carlosf.lugo@urosario.edu.co (C.L.-S.); 2Unidad de Extensión Hospitalaria, Hospital Universitario Mayor Méderi, Bogota 111411, Colombia; 3Studies in Translational Microbiology and Emerging Diseases (MICROS) Research Group, School of Medicine and Health Sciences, Universidad de Rosario, Bogota 111221, Colombia

**Keywords:** mycosis, rare yeasts, Latin America, invasive fungal infection, *Trichosporon*, *Rhodotorula*, *Geotrichum*, *Saccharomyces*

## Abstract

Systemic infections caused by rare yeasts are increasing given the rise in immunocompromised or seriously ill patients. Even though globally, the clinical significance of these emerging opportunistic yeasts is increasingly being recognized, less is known about the epidemiology of rare yeasts in Latin America. This review collects, analyzes, and contributes demographic and clinical data from 495 cases of infection caused by rare yeasts in the region. Among all cases, 32 species of rare yeasts, distributed in 12 genera, have been reported in 8 Latin American countries, with *Trichosporon asahii* (49.5%), *Rhodotorula mucilaginosa* (11.1%), and *Saccharomyces cerevisiae* (7.8%) the most common species found. Patients were mostly male (58.3%), from neonates to 84 years of age. Statistically, surgery and antibiotic use were associated with higher rates of *Trichosporon* infections, while central venous catheter, leukemia, and cancer were associated with higher rates of *Rhodotorula* infections. From all cases, fungemia was the predominant diagnosis (50.3%). Patients were mostly treated with amphotericin B (58.7%). Crude mortality was 40.8%, with a higher risk of death from fungemia and *T. asahii* infections. Culture was the main diagnostic methodology. Antifungal resistance to one or more drugs was reported in various species of rare yeasts.

## 1. Introduction

The increasing incidence of severely ill and immunocompromised patients is leading to the rise in invasive fungal infections caused by rare yeasts, which comprise non-*Candida* and non-*Cryptococcus* species [1,2]. Even though these microorganisms are not rare per se, as they abound in the human microbiome, the environment, and food, they are called rare as they hardly ever cause disease in humans, mostly because of their low pathogenicity [3]. Patients at risk include those in supportive and intensive care, people with cancer or hematological malignancies, solid organ and stem cell transplant recipients, patients on active treatment with high-dose corticosteroids, among others [4]. Hence, the weakened immune status of the host is a crucial determinant of the severity of the disease these rare yeasts cause, which in many cases results in high mortality rates, particularly in patients with disseminated infection [4,5]. The outcome for these infections is also determined by the etiologic agent, since various species of rare yeasts, including nonpathogenic, are characterized by intrinsic or acquired resistance to one or more commonly used antifungal drugs, which reduces treatment options and worsens the prognosis [6,7].

Although rare yeast pathogens cause less than 2% of cases of fungemia and invasive infections, as reported by various surveillance studies from different geographic regions, these yeasts are typically encountered in health care-associated settings [8,9,10]. This scenario leads to longer hospital stays as well as additional therapies, which significantly increases patient morbidity and mortality. Together, this represents a great financial impact on hospital resources, outweighing the burden of the underlying diseases alone and hampering medical advancements [11,12]. Moreover, the identification of rare yeasts remains a challenge and the diagnosis depends on a high degree of suspicion. As such, many infections caused by these infrequent pathogens are still misdiagnosed or have late diagnosis, leading to inadequate treatment and, therefore, poorer outcomes [4].

Globally, the importance of rare yeasts as emerging opportunistic human pathogens is increasingly being recognized; thus, clinical guidelines for their diagnosis and management are already available [1,6]. However, less is known about the infections that are caused by species of rare yeasts in Latin America and knowledge about distinctive epidemiological traits of these pathogens in the region is scarce and insufficient. Therefore, this review aims to collect and analyze the demographic and clinical characteristics of a large set of patients affected by rare yeasts from Latin American countries. The information herein presented adds to local data that can be useful for identifying patients at the highest risk of developing severe infections caused by rare yeasts and puts together data on antifungal susceptibility testing of these species, which can guide choosing antifungal therapy that results in better outcomes. In addition, this review seeks to raise awareness of the significance of these mycoses, a growing global public health problem, and of the importance of reporting clinical cases and surveys on rare yeasts in the region to increase our understanding of these life-threatening pathogens. Regional epidemiological studies, surveys, and data collections are essential to inform local differences in the prevalence of medically important fungi, as well as to highlight diagnostic and management priorities.

## 2. Materials and Methods

### 2.1. Data Collection

A literature search comprising studies published until March 2022 was undertaken, focusing on case reports and surveys of rare yeasts causing infection in Latin American countries. Rare yeasts were defined as non-candidal and non-cryptococcal species, according to the European Confederation for Medical Mycology (ECMM), the International Society for Human and Animal Mycology (ISHAM), and the American Society for Microbiology (ASM) and include the yeasts that are most well described in clinical settings [1]. Publications in English, Spanish, and Portuguese from PubMed, SciELO, and Google databases were included. The literature search was based on the keywords “invasive”, “infection”, “case”, “report”, in combination with the names of the genera *Geotrichum* (*Dipodascus* or *Galactomyces*), *Kodamaea* (*Pichia*), *Magnusiomyces*, *Malassezia* (*Pityrosporum*), *Pseudozyma* (*Moesziomyces* or *Dirkmeia*), *Rhodotorula* (*Cystobasidium*), *Saccharomyces*, *Saprochaete*, *Sporobolomyces,* and *Trichosporon*, and each of the 21 countries from Latin America, namely, Argentina, Bolivia, Brazil, Chile, Colombia, Costa Rica, Cuba, Dominican Republic, Ecuador, El Salvador, Guatemala, Haiti, Honduras, Mexico, Nicaragua, Panama, Paraguay, Peru, Puerto Rico, Uruguay, and Venezuela. Additional fungal names were used in the search, where taxonomic changes had occurred or synonyms were in use [1,3].

All cases of fungemia and other invasive fungal infections, as defined by the European Organization for Research and Treatment of Cancer and the Mycoses Study Group [13], as well as cases of urinary tract infections and catheter-related infections, were included when stated. In immunocompromised patients or those with other underlying conditions, cases of oral infection, vaginal infection, and folliculitis were included, as they represent distinct cases. Superficial fungal infections of the skin, nails, hair, and mucosal surfaces were not included in this review. Epidemiological data of patients with rare yeast infections, including age, sex, risk factors, or underlying conditions, treatment, and outcome, together with data on the site of the body affected, the diagnostic methods utilized, and antifungal susceptibility testing of the yeasts were collected when available.

### 2.2. Statistical Analysis

Data are shown as numbers and percentages. For quantitative variables, median and range are shown. A multivariate logistic regression was performed to assess the relation between the genera of the etiological agent and the variables, (i) type of infection caused and (ii) risk factor or underlying condition of affected patients. A multivariate logistic regression was performed, in addition, to assess the relation between outcome and the explanatory variables, (i) age, (ii) sex, (iii) type of infection, and (iv) species of the etiological agent. Data were checked for multicollinearity with the Belsley–Kuh–Welsch technique. Heteroskedasticity and normality of residuals were assessed, respectively, by the White test and the Shapiro–Wilk test. Adjusted odds ratio (OR) and 95% confidence interval (CI) are presented. The association between outcome and the explanatory variables, (i) risk factor or underlying condition, (ii) type of infection, (ii) treatment, and iv) genera of the etiological agent, was tested with the Chi-squared test. The alpha risk was set to 5% (α = 0.05). Statistical analysis was performed with the online platform EasyMedStat (version 3.9; www.easymedstat.com (accessed on 18 May 2023)) and the software GraphPad Prism version 9. The population of each Latin American country, to calculate the number of cases of rare yeast infections per million residents per country, was obtained from www.worldometers.info (accessed on 2 May 2023) [1,14].

## 3. Results

### 3.1. Species of Rare Yeasts

To date, 96 publications reporting 495 cases of infections caused by rare yeasts have been published in Latin America. From the cases, dating back to 1997 [15,16,17,18], 288 (58.2%) were reported from Brazil, followed by Argentina with 110 cases (22.2%), Mexico with 63 (12.7%), Venezuela with 12 (2.4%), Chile with 11 (2.2%), Colombia with 8 (1.6%), Cuba with 2 (0.4%), and Guatemala with 1 case (0.2%). Cases of severe infections caused by rare yeasts from Bolivia, Costa Rica, Dominican Republic, Ecuador, El Salvador, Haiti, Honduras, Nicaragua, Panama, Paraguay, Peru, Puerto Rico, and Uruguay have not been reported until now (Figure 1).

From the species of rare yeasts, 396 (80%) were identified to the species level, accounting for 32 species, distributed in 12 genera: *Geotrichum candidum* [19,20,21], *Kloeckera apiculata* (*Hanseniaspora uvarum*) [17], *Kodamaea ohmeri* [22,23,24,25], *Magnusiomyces capitatus* (*Geotrichum capitatum*, *Saprochaete capitata*, *Blastoschizomyces capitatus*) [26,27,28,29,30], *Malassezia furfur* [18,31,32,33,34,35], *Malassezia pachydermatis* [36], *Malassezia sympodialis* [34,37,38], *Millerozyma farinosa* (*Pichia farinosa*) [39], *Pichia angusta* (*Hansenula polymorpha*) [40], *Pichia anomala* (*Wickerhamomyces anomalus*) [40,41], *Pseudozyma aphidis* [42,43], *Rhodotorula dairenensis* [44], *Rhodotorula glutinis* [15,34,39,45,46], *Rhodotorula minuta* (*Cystobasidium minutum*) [44,47], *Rhodotorula mucilaginosa* (*Rhodotorula rubra*) [40,44,46,48,49,50,51,52,53], *Rhodotorula toruloides* [44], *Saccharomyces cerevisiae* [18,21,54,55,56,57,58,59,60,61,62,63,64,65], *Saprochaete clavata* (*Geotrichum clavatum*) [66], *Trichosporon asahii* [40,49,66,67,68,69,70,71,72,73,74,75,76,77,78,79,80,81,82,83,84,85,86,87,88,89,90,91,92,93], *Trichosporon asteroides* [76,91], *Trichosporon beigelii* [17,18], *Trichosporon coremiiforme* [71,76,94], *Trichosporon cutaneum* [95], *Trichosporon dermatis* [71,76], *Trichosporon faecale* [71,91], *Trichosporon figueirae* [18], *Trichosporon inkin* [16,67,71,74,79,91,96,97,98,99,100,101], *Trichosporon japonicum*, *Trichosporon jirovecii* [71], *Trichosporon montevideense* [79], *Trichosporon mucoides* [23,57,102,103], and *Trichosporon ovoides* [71].

In the remaining 99 cases (20%), species identification was not possible; however, the genera, *Geotrichum* [18,30], *Malassezia* [104,105,106], *Rhodotorula* [21,46,107,108,109], *Saccharomyces* [109], or *Trichosporon* [21,39,62,79,91,109,110], were identified.

Among the 396 cases in which the genus and species were identified, *T. asahii* was the most common species found, with 196 reports (49.5%), followed by *R. mucilaginosa* with 44 cases (11.1%), *S. cerevisiae* with 31 cases (7.8%), *M. furfur* and *T. inkin* with 20 cases each (5.1%), *G. candidum* with 14 cases (3.5%), and *T. asteroides* with 11 cases (2.8%). The other 25 species, accounting for 15.2% of cases, were reported with less than 10 cases each (Figure 2). From the 32 species reported in Latin America, 22 were identified in Brazil, 20 in Argentina, 6 in Mexico, 5 each in Chile and Colombia, 2 in Cuba, and 1 each in Venezuela and Guatemala. While *T. asahii* was reported in Argentina, Brazil, Chile, Colombia, Guatemala, and Mexico, *R. mucilaginosa* was reported only in Brazil, Chile, and Colombia and *S. cerevisiae* only in Argentina, Brazil, and Chile.

### 3.2. Clinical Characteristics of Patients

Documented demographic and clinical characteristics of patients with infections caused by rare yeasts in Latin America are summarized in Table 1. From the patients with data on sex, 144 (58.3%) were men and 103 (41.7%) were women. From the 179 patients with data on age, cases were described from neonates to 84-year-old patients, with an average age of 34.7 years and median 39.5 years. Neonates included 10 newborns with less than 24 h of life and 5 infants between 9 and 22 days of age.

In 312 cases, underlying conditions or risk factors were documented, with leukemia, central venous catheter (CVC), cancer, surgery, and antibiotic use the most commonly reported among patients (Table 1). While some patients presented two or more underlying conditions or risk factors, most patients (70%) had only one. In multivariate analysis, surgery (OR:12.11; CI: 2.57–57.02, *p* = 0.0016) and antibiotic use (OR: 168.57; CI: 5.52–5149.53; *p* = 0.0033) were associated with higher rates of *Trichosporon* infections. In addition, CVC (OR: 4.51; CI: 1.74–11.68, *p* = 0.0019), leukemia (OR:4.76; CI: 2.63–8.63], *p* < 0.0001), and cancer (OR: 5.41; CI: 2.56–11.45; *p* < 0.0001) were associated with higher rates of *Rhodotorula* infections. No other association was found between underlying conditions or risk factors and the etiological agent.

From all 495 cases, fungemia alone accounted for 249 cases (50.3%), followed by 86 cases (17.4%) of other invasive infections and 81 cases (16.4%) of urinary tract infection. The remaining 79 cases (16%) were distributed among other various infections, including abscesses, folliculitis, endophthalmitis, and oral, oropharyngeal, and vaginal infections. With respect to the etiological agent, both *Trichosporon* and *Rhodotorula* infections were more likely to be fungemia (OR: 2.28; CI: 1.57–3.31 and OR: 0.061; CI: 0.029–0.13, respectively, with *p* < 0.001), while cases of *Geotrichum* (OR: 8.63; CI: 1.96–37.94; *p* < 0.001) and other genera (OR: 2.09; CI: 1.09–4.01; *p* = 0.035) were more likely to be infections other than fungemia, invasive infection, or urinary tract infection.

Treatment was reported in 213 patients. Of those, 130 (61.3%) received monotherapy either with amphotericin B (29.2%), fluconazole (19.3%), voriconazole (11.3%), or itraconazole (1.4%). In 62 patients (29.2%), combined therapy, mainly amphotericin B plus an azole drug, was reported. Five (2.4%) patients received treatment, but this was not specified. Fifteen (7.1%) patients were reported not to have received any antifungal treatment.

The outcome for the patients was reported in 277 cases. Of those, 164 (59.2%) survived and 113 (40.8%) died. Multivariate analysis revealed that per every year of life, patients were 1.02 times more likely to die. Additionally, patients who died were 13.6 years older than those who survived (*p* < 0.001) (Table 2). Patients’ sex was not associated with the outcome. In Table 2, reference levels, defined automatically by the statistical program, serve as “baseline” values for a given variable, and helps comparing the data generated.

Among patients with data on the outcome, there was a statistically significant higher risk to die of fungemia than of other infections (OR: 23.59; CI: 3.1–179.78; *p* = 0.00229) and *T. asahii* infections were found more likely to cause death than infections caused by other species of rare yeasts (OR: 11.28; CI: 5.61–22.66; *p* < 0.0001). Univariate analysis revealed that patients with leukemia, corticosteroid use, bacteremia and mechanical ventilation were also statistically more likely to die than patients with other risk factors or underlying conditions, while patients with surgery were more likely to survive (Table 3). Generally, fungemia and invasive infection as well as infections caused by species of the genus *Trichosporon* were found more likely to cause death, while patients with infections caused by species of the genera *Rhodotorula* and *Geotrichum* were more likely to survive. The outcome was not associated with the treatment received.

### 3.3. Diagnosis

In 487 out of the 495 patients from Latin America with infections caused by rare yeasts, the method used for diagnosis was specified. Of these, cultures from diverse biological samples, including blood, respiratory tract samples, urine, abdominal/thoracic fluids, and biopsies, among others, were done in most cases (95.9%). Biochemical identification, following culture, was performed in 190 cases (39.3%), using automated systems based on carbohydrate and other compound assimilation tests (e.g., Vitek, the API 20C, and the ID 32C yeast identification systems from bioMérieux, Inc. (Marcy-l’Étoile, France)). Molecular identification was performed in 201 cases (41.4%); 90 from cultures and 111 directly from patients’ specimens. Molecular techniques included sequencing of the internal transcribed spacer (ITS) region, the D1/D2 region, and other species-specific genes, as well as genotyping of the isolates to further characterize certain species. Microscopic examination was performed in 77 cases (15.8%), 58 following culture and 19 by observing fungal elements directly in clinical samples. Matrix-assisted laser desorption/ionization time-of-flight mass spectrometry (MALDI-TOF MS) was used for species identification in 28 cases (5.8%), when the pure culture of the yeasts was available. Of these yeasts identified by MALDI-TOF MS, 17 were *R. mucilaginosa*, 10 were other species of *Rhodotorula,* and 1 was *T. inkin*.

### 3.4. Antifungal Susceptibility

The susceptibility to commonly used antifungal drugs was determined in 164 (33.1%) isolates of rare yeast from Latin America. In 79 (48.2%) of them, the antifungal susceptibility testing was based on the European Committee on Antimicrobial Susceptibility Testing-Subcommittee on Antifungal Susceptibility Testing (EUCAST-AFST) method E.Def.7.2. In 70 isolates (42.7%), the microdilution method in RPMI broth, according to the M27-A3 guideline of the Clinical and Laboratory Standards Institute (CLSI) [111], was used. Other methodologies, such as E-test strips (bioMerieux, Inc., Marcy-l’Étoile, France), disk diffusion in accordance with the CLSI protocol M44-A2, the Vitek-2 Compact system (bioMerieux, Inc., Marcy-l’Étoile France), and Fungifast^®®^ AFG (ELITech Microbiology, Puteaux, France), were reported in the remaining 15 isolates (9.1%).

From the isolates with antifungal susceptibility data, 129 (78.7%) belonged to the genus *Trichosporon*, followed by 18 (11%) *Saccharomyces* isolates, 5 (3%) of *Rhodotorula,* and 12 (7.3%) of other genera. Even though minimum inhibitory concentrations (MIC) data for one or more antifungal drugs were reported for all isolates, interpretation for susceptibility or resistance could not always be established. Nevertheless, resistance to fluconazole was reported in 27 (16.5%) isolates; 10 of them were *T. asahii*, 3 each of *S. cerevisiae* and *Trichosporon* sp., 2 each of *R. mucilaginosa* and *Rhodotorula* sp., and 1 each of *G. candidum*, *K. ohmeri*, *M. capitatus*, *M. sympodialis*, *R. glutinis,* and *T. inkin*. Resistance to 5-fluorocytocine was reported in 25 isolates, of which, 19 were *T. asahii* (76%), 5 were *S. cerevisiae* (20%), and 1 were *T. inkin* (4%). Resistance to amphotericin B was reported in 17 isolates of *T. asahii*, 2 of *Trichosporon* spp., and 1 each of *M. sympodialis* and *T. inkin*. Itraconazole resistance was reported in two isolates of *T. asahii* and one each of *R. mucilaginosa* and *R. glutinis*, while only two isolates of *T. asahii* were reported as voriconazole resistant. Concomitant resistance to amphotericin B and fluconazole was reported in two isolates each of *T. asahii* and *Trichosporon* spp., and one isolate of *M. sympodialis*. Four isolates of *T. asahii* were additionally concomitantly resistant to 5-fluorocytocine and two other isolates of *T. asahii* were, in addition, resistant to itraconazole and voriconazole, making them pan-drug resistant. Concomitant resistance to fluconazole and itraconazole was reported in one isolate each of *R. mucilaginosa* and *R. glutinis* and concomitant resistance to amphotericin B and 5-fluorocytocine was found in seven isolates of *T. asahii*.

## 4. Discussion

In the growing population of patients receiving medical interventions and those with a weakened immune system, infections caused by rare yeasts have emerged as an important health concern, given that these infections are usually associated with significant morbidity and high mortality rates [1,6]. The first difficulty when fighting these emerging infectious pathogens is that little is known about their modes of transmission, how to best diagnose them, and how to treat them, particularly when it is unknown if they present resistance or reduced susceptibility to certain antifungal drugs. By analyzing 495 cases of rare yeast infections from 8 Latin American countries, this review is contributing to an increase in our knowledge on the demographic and clinical characteristics of patients at risk as well as on the diagnostic methodologies and antifungal susceptibility of the etiological agents.

Similar to the global epidemiology of rare yeast infections [1,2], *T. asahii* and *R. mucilaginosa* were found to be the two most common species of rare yeasts isolated from clinical specimens in Latin America. However, in this region, infections caused by *Saccharomyces*, *Malassezia*, other species of *Trichosporon* and *Rhodotorula*, and *G. candidum* prevail over cases by *Kodamea*, which is the third most common genus of rare yeast causing invasive infections in hospitalized patients worldwide. In Latin American countries, *Kodamea* species could be misidentified as *Candida* species, due to their phenotypic resemblance and because molecular diagnostic methods, which are required for proper identification, are not always available [3].

Notably, this review complements the epidemiology of non-*Candida* and non-*Cryptococcus* invasive infections in Latin America and the world by adding data on reports of rare yeast species that were not included in the latest guidelines [1]. Specifically, the isolation of *M. capitatus* (*S. capitata*) in Argentina, *R. mucilaginosa* in Colombia, *G. candidum* and *R. glutinis* in Mexico, and *Malassezia* species in Argentina, Brazil, Colombia, Mexico, and Venezuela. In addition, we include for the first time the reports on the isolation of *M. farinosa*, *P. angusta*, *P. anomala*, *R. dairenensis,* and *R. toruloides* from human samples in Brazil [39,40,41,44] and *K. apiculata* in Cuba [17]. This last species being rarely recovered from clinical samples [112,113]. Here, we also confirm that *Sporobolomyces* species have not been reported in Latin America.

As previously reported in global surveys [4,6], rare yeast infections in Latin America also occur similarly among male and female patients of all ages. However, considering the outcome, our review revealed that patient at an older age are less likely to survive, probably because aging is associated with declining immunity and increased vulnerability to other malignancies [114,115]. The prevalence and outcome of rare yeast infections in Latin America varied as well depending on the risk factors of patients and the etiological agent, even though there are slight differences compared to the global epidemiology of these infections. While hematologic malignancies and neutropenia are considered classical risk factors for invasive *Trichosporon* infection [2,116,117], in Latin America, surgery and antibiotic use prevail among patients affected by species of this genus. Conversely, CVC, leukemia, and cancer were associated with higher rates of *Rhodotorula* infections, which agrees with the typical risk factors for this genus [5,46]. Mortality associated with invasive infections caused by *T. asahii* is generally higher than that of *Candida* infections [117,118]. In Latin America, about 80% of patients affected by *T. asahii* died. In addition, a larger proportion of patients with corticosteroid use (78.9%), bacteremia (63.6%), and leukemia (61.5%) died compared to patients with other risk factors. The recognition of the underlying conditions and etiological agent is, therefore, very important, as outcomes for patients are often worse because of them [5].

Rare yeasts most commonly cause fungemia, as found in this study. Nevertheless, non-blood stream infections (BSI) caused by these pathogens accounted for almost half the cases, including catheter-related infection, urinary tract infection, pneumonia, peritonitis, meningoencephalitis, and endocarditis. As reported in the literature, *Trichosporon* and *Rhodotorula* invasive infections, particularly by *T. asahii* and *R. mucilaginosa*, are more likely to be fungemia [2,5,11,46,117], while other genera can cause diverse clinical manifestations, such as *Geotrichum* causing pulmonary infection or *Saccharomyces* causing oral infection [30,60,119,120].

Although the outcome for patients in Latin America was not associated with the treatment they received, it is important to consider the therapeutic options when fighting rare opportunistic infections. In the hospital setting, prophylactic antifungal therapy is mostly prescribed upon suspicion of candidemia, with echinocandins being the first option for *Candida* species [121]. In fact, the use of echinocandins is strongly recommended in patients with a high risk of serious sepsis or septic shock, due to their demonstrated efficacy and broad spectrum of action [122]. However, this family of antifungals is ineffective in the treatment of species that present natural or intrinsic resistance, including the basidiomycetes *Trichosporon*, *Rhodotorula,* and *Malassezia* [48,120,123] that are the most common genera of rare yeasts affecting critically ill patients. This explains why patients receiving caspofungin as empirical treatment are susceptible to breakthrough infections and why after diagnosis, amphotericin B and fluconazole are the most common antifungal therapy.

Apart from the intrinsic resistance to echinocandins of *Trichosporon* and *Rhodotorula*, some isolates of these two genera, mainly *T. asahii* isolates, have been reported in Latin America to be resistant to fluconazole, itraconazole, 5-fluorocytocine, and amphotericin B, separately or concomitantly. This clearly leaves physicians with an even smaller arsenal of therapeutic options against these pathogens. Moreover, while voriconazole exhibits the best in vitro activity against *T. asahii* clinical isolates and resistance is uncommon [97,124], our study found that in Latin American countries, this triazole is not considered as first-line therapy. Fluconazole resistance has also been identified in *M. sympodialis* and ascomycetes yeasts such as *S. cerevisiae*, *G. candidum*, *K. ohmeri,* and *M. capitatus* from Latin America. In Europe, *S. cerevisiae* and *G. candidum*, particularly, have been reported to be resistant to many current antifungals including echinocandins [125]. Together, these findings highlight the importance of identifying the genus and species of the etiological agents and applying antifungal susceptibility testing on all rare yeasts. In Latin America, antifungal susceptibility was determined in less than a third of the isolates. In addition, it is necessary to further investigate whether in vitro results are predictors of the clinical response to antifungal therapy.

Regarding the diagnosis of rare yeast infections, we show that in Latin American countries, culture is the preferred and most accessible approach that can be utilized to identify the etiological agents. Globally, cultures from clinical samples remain the gold standard for the diagnosis of fungal infections, allowing, moreover, for susceptibility testing [1,126]. However, a final and accurate identification by biochemical, proteomic, and molecular methods is generally limited to commercial platforms that are usually directed toward the identification of common rather than rare fungi. This is why the use of MALDI-TOF MS to identify non-*Candida* and non-*Cryptococcus* yeast species has been found to be considerably less suitable, given the absence of reference spectra for other genera [127]. In Latin America, particularly, this technique has only been used to identify *Rhodotorula* species. Microscopy and histopathology, utilizing fungal-specific stains, can be suggestive and aids the diagnosis, but these approaches depend on the skills and experience of laboratory personnel and cannot be used alone [6]. In Latin America, more advanced diagnostic approaches and methodologies, both culture- and non-culture-based, are still needed to meet the requirements to diagnose fungal infections caused by rare yeasts, especially considering that many countries lack sufficient resources for testing.

It is important to note as well that several species of rare yeasts are medically important in Latin America, even though cases of severe infections have not been reported in many countries. *S. cerevisiae*, *Rhodotorula* sp., and *T. cutaneum* were found as part of the vaginal yeast flora, some of them causing vulvovaginitis, in pregnant women living in Peru [128]. *Trichosporon* and *Rhodotorula* species have been reported as causing vaginosis in Venezuela [129]. *Malassezia* species have also been reported in cases of pityriasis versicolor in infants under one year of age in Dominican Republic [130] and *M. globosa*, *M. furfur*, *M. sympodialis,* and *Malassezia slooffiae* causing pityriasis versicolor in more than 100 patients from 1 month to 63 years of age in Paraguay [131]. In Puerto Rico, *Malassezia* and *Saccharomyces* were found in the fungal cervicovaginal biota in women with human papillomavirus (HPV) infection [132]. *Trichosporon* species causing invasive and superficial infections and *Saccharomyces* causing fungemia following probiotic ingestion have been associated with overall mortality in Brazil [124,133]. These reports highlight the importance of increasing awareness among physicians and laboratory personnel for better diagnosing rare yeast infections, especially in the early stages of infection, which is essential to avoid further complications.

## 5. Conclusions

The epidemiology of rare yeast infections differs between regions, countries, and even between institutions. Not only is the distribution of these pathogens in the environment or in the human microbiota different, but also the number and spectrum of susceptible patients and the availability of diagnostic tools, which causes species of rare yeasts to be misidentified and the prevalence of their infections to be underestimated. Because of the significant medical impact of these yeasts, timely diagnosis and accurate patient care are essential. That, in turn, relies on recognizing disease patterns and having access to appropriate therapeutic options. During the last decade, the array of rare yeasts that are being recognized as human opportunistic pathogens has expanded, because of the advances in health care and improved diagnostic techniques. However, regardless of this progress, some species of rare yeasts are still difficult to identify by phenotypic methods and many species show important resistance profiles to commonly used antifungals. Clinicians and microbiologists must, therefore, be aware of and stay alert for the appearance of these emerging pathogens as they can lead to serious complications that aggravate the underlying diseases of patients at risk and negatively impact patient care. Epidemiological surveillance is key to obtaining a true picture of the burden of rare yeast infections and to identify local unmet needs, which ultimately will be needed to optimize the clinical management of patients.

## Figures and Tables

**Figure 1 jof-09-00747-f001:**
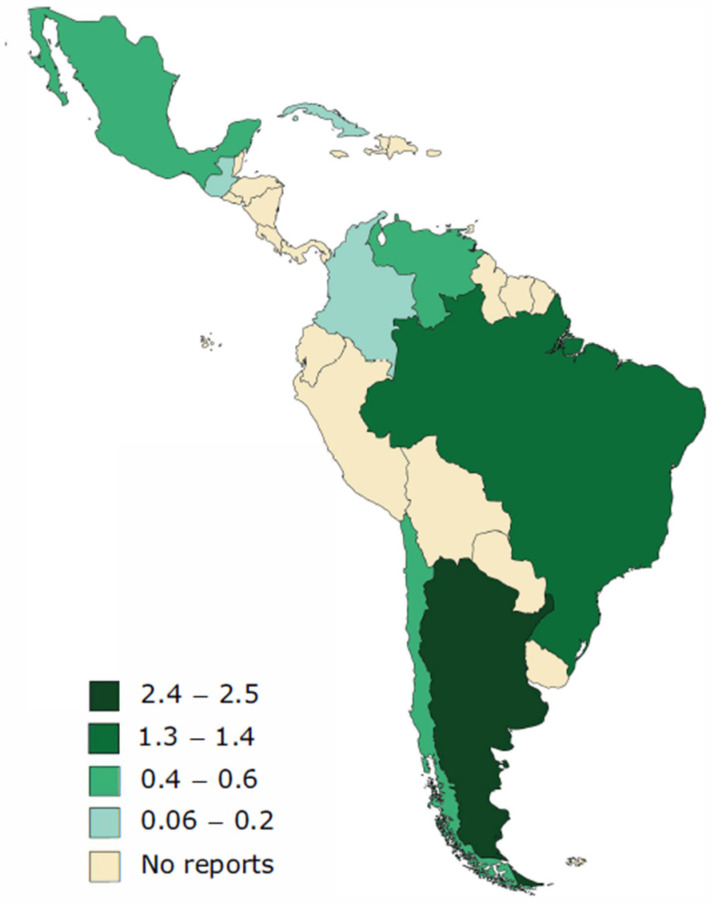
Number of cases of rare yeast infections, per million inhabitants, reported in eight Latin American countries between 1997 and 2022.

**Figure 2 jof-09-00747-f002:**
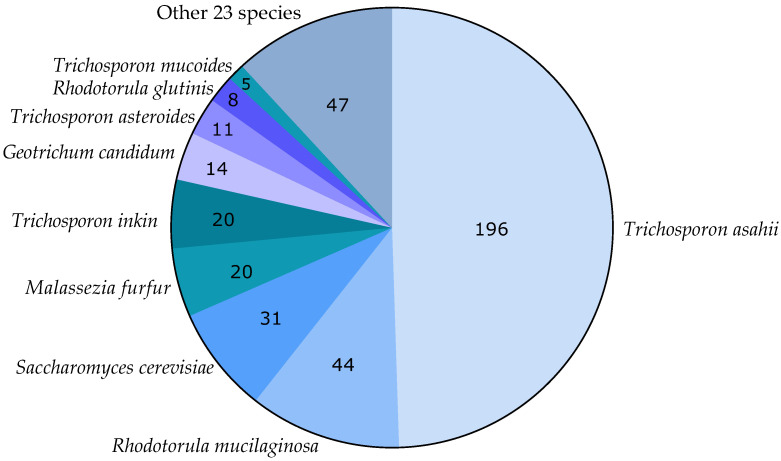
Distribution of species of rare yeasts reported to cause infection in Latin America.

**Table 1 jof-09-00747-t001:** Demographic and clinical characteristics of patients from Latin America diagnosed with infections caused by rare yeasts.

Characteristic of Patients	Number (%)
Sex (*n* = 247)	
Male	144 (58.3)
Female	103 (41.7)
Underlying condition (*n* = 312)	
Leukemia	78 (25.0)
Central venous catheter	58 (18.6)
Cancer	48 (15.4)
Surgery	36 (11.5)
Antibiotic use	28 (9.0)
Mechanical ventilation	26 (8.3)
Solid organ transplantation	23 (7.4)
Corticosteroid use	21 (6.7)
Diabetes mellitus	21 (6.7)
Human immunodeficiency virus	17 (5.4)
Chemotherapy	14 (4.5)
Hematopoietic stem cell transplant	14 (4.5)
Urinary catheter	14 (4.5)
COVID-19	9 (2.9)
Premature birth	9 (2.9)
Other	110 (35.3)
Diagnosis (*n* = 495)	
Fungemia	249 (50.3)
Invasive infection	86 (17.4)
Urinary tract infection	81 (16.4)
Other infections	79 (16.0)
Treatment ^1^ (*n* = 213)	
Amphotericin B	125 (58.7)
Fluconazole	65 (30.5)
Voriconazole	40 (18.8)
Others	19 (8.9)
None	15 (7.0)
Outcome (*n* = 277)	
Alive	164 (59.2)
Dead	113 (40.8)

^1^ Usage of the antifungal alone or in combination.

**Table 2 jof-09-00747-t002:** Multivariate analysis of characteristics associated with death among patients from Latin America diagnosed with infections caused by rare yeasts (*n* = 277).

		Survived *n* = 164	Died *n* = 113	OR (95% CI)	*p*-Value
Sex	Male	63 (22.7%)	39 (14.1%)	0.792 [0.425–1.48]	0.463
	Female	52 (18.8%)	25 (9.0%)	Reference	
	Missing data	49 (17.7%)	49 (17.7%)		
Age	With data	101 (36.5%)	57 (20.6%)	-	-
	Missing data	63 (22.7%)	56 (20.2%)	-	-
	Average in years (SD)	30.38 (±23.37)	43.94 (±25.18)	-	**<0.001**
	Risk for each 1-unit increase			1.02 [1.01–1.04]	**0.0013**
Diagnosis	Fungemia	78 (28.2%)	92 (33.2%)	23.59 [3.1–179.78]	**0.00229**
	Invasive infection	56 (20.2%)	12 (4.3%)	4.36 [0.533–35.75]	0.17
	Urinary tract infection	10 (3.6%)	8 (2.9%)	16 [1.75–146.31]	**0.0141**
	Other	20 (7.2%)	1 (0.4%)	Reference	
Species/ Genus	*T. asahii*	21 (7.6%)	74 (26.7%)	11.28 [5.61–22.66]	**<0.0001**
	*R. mucilaginosa*	17 (6.1%)	10 (3.6%)	2 [0.784–5.1]	0.147
	*Rhodotorula*	27 (9.7%)	5 (1.8%)	0.593 [0.202–1.74]	0.342
	*Trichosporon*	35 (12.6%)	4 (1.4%)	0.366 [0.116–1.15]	0.0864
	Other	64 (23.1%)	20 (7.2%)	Reference	

^1^ Folliculitis, oral and oropharyngeal infections.

**Table 3 jof-09-00747-t003:** Univariate analysis of characteristics associated with death among patients from Latin America diagnosed with infections caused by rare yeasts (*n* = 277).

		Survived *n* = 164	Died *n* = 113	OR (95% CI)	*p*-Value
**Underlying** **condition**	Leukemia	30	48	3.92 [1.65–9.33]	**0.00204**
	CVC	35	22	0.754 [0.317–1.8]	0.524
	Cancer	32	15	1.33 [0.521–3.4]	0.55
	Surgery	30	5	0.262 [0.0842–0.814]	**0.0206**
	Mechanical ventilation	8	16	6.32 [0.887–45.04]	0.0657
	Antibiotic use	9	14	0.398 [0.0356–4.45]	0.455
	SOT	12	7	0.993 [0.306–3.22]	0.991
	Corticosteroid use	4	15	9.01 [1.55–52.29]	**0.0143**
	Diabetes mellitus	12	8	0.878 [0.251–3.07]	0.838
	Chemotherapy	7	7	0.511 [0.124–2.1]	0.352
	HSCT	11	3	0.203 [0.0411–1.01]	0.0511
	Urinary catheter	5	7	0.555 [0.0576–5.36]	0.611
	Bacteremia	4	7	6.58 [1.54–28.04]	**0.0108**
	Other	54	33	0.946 [0.444–2.02]	0.885
**Diagnosis**	Fungemia	78	92	4.77 [2.71–8.4]	**<0.001**
	Invasive infection	56	12	0.23 [0.12; 0.46]	**<0.001**
	Urinary tract infection	10	8	1.17 [0.45; 3.05]	0.807
**Treatment**	Amphotericin B	71	54	1.19 [0.735–1.94]	0.475
	Fluconazole	47	28	0.841 [0.481–1.47]	0.544
	Voriconazole	22	18	1.19 [0.593–2.38]	0.628
**Genus**	*Trichosporon*	79	86	3.39 [1.99–5.76]	**<0.001**
	*Rhodotorula*	51	16	0.37 [0.2–0.7]	**0.003**
	*Geotrichum*	15	1	0.088 [0.011–0.68]	**0.003**
	*Saccharomyces*	9	5	0.79 [0.26–2.43]	0.785

CVC: Central venous catheter; SOT: Solid organ transplantation; HSCT: Hematopoietic stem cell transplant.

## Data Availability

All data supporting this review are from previously reported studies and datasets, which have been properly cited. The processed data are available from the corresponding author upon request.

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
