# Peer review of "Rare Yeasts in Latin America: Uncommon Yet Meaningful"

_jof, 2023, doi:10.3390/jof9070747_

Round 1

Reviewer 1 Report

In their manuscript entitled, “Rare yeasts in Latin America: uncommon yet meaningful”, Gil and colleagues perform a systematic review of the literature surrounding cases of systemic fungal infections caused by rare and/or neglected yeasts, not including Candida or Cryptococcus infections. The authors do an excellent job cataloging the taxonomic diversity of rare yeast infections throughout Latin America, and provide a thorough breakdown of patient demographics, comorbidities, treatments, and antifungal resistance. Overall, the article is well written and provides useful information for clinicians and scientists interested in emerging fungal infections          in Central and South America. There are a few minor grammatical and stylistic issues that need addressing prior to publication. These issues are outlined below.

Minor Comments

1.     Line 14: Delete the comma following “globally”.

2.     Line 16: Add a comma following “analyses” and delete “with” preceding “demographic”.

3.     Line 18: Change “being” to “with” at end of line.

4.     Line 20: Change “men” to “male”.

5.     Line 24: Change “to die of” to “of death from”.

6.     Line 33: Delete commas surrounding “per se”.

7.     Line 34: Delete “so” preceding “called”.

8.     Line 40: Delete commas surrounding “in many cases”.

9.     Line 42: Change sentence to read: “…of rare yeasts (including non-pathogenic)…”

10.  Line 44: Change “…drugs, which…” to “…drugs that…”.

11.  Line 46: Delete comma following “infections”.

12.  Line 48: Change “…therapies, which…” to “…therapies that…”.

13.  Line 52: Change “…diagnosis depends on…” to “…diagnosis results in…”.

14.  Line 59: Delete comma following “region”.

15.  Line 60: Change “aimed” to “aims”.

16.  Line 92: Insert “those” preceding “…with other underlying…”.

17.  Line 93: Delete “in addition”.

18.  Line 98: Delete comma following “yeasts”.

19.  Line 171: Change “being” to “with”.

20.  Line 187: Change sentence to read “…more likely to be infections other than…”.

21.  Line 188: Insert comma following “invasive infection”.

22.  Line 205: Replace “Opposite” with “In contrast” and delete “statistically”.

23.  Line 206: Change sentence to read: “…significantly higher odds of survival with urinary…”.

24.  Line 207: Remove the “s” from “corticosteroids” to make it singular.

25.  Line 219: Add “s” to “culture” to make it plural.

26.  Line 220: Insert “and” preceding “biopsies”.

27.  Line 221: Replace commas with parentheses surrounding “among others”.

28.  Line 222: Delete “, by” preceding “using”.

29.  Lines 223 – 224: Change sentence to read: “…assimilation tests (e.g. Vivek, API 20C, and ID 32C yeast identification systems from bioMerieux, Inc.).”

30.  Line 225: Change comma to semicolon preceding “90”.

31.  Line 229: Delete “just”.

32.  Line 230: Insert “mass spectrometry” preceding “(MALDI-TOF)”.

33.  Line 231: Change “always after” to “when”.

34.  Line 241: Delete comma following reference 111.

35.  Line 243: Insert comma after “…France)”.

36.  Line 248: Delete comma after “established”.

37.  Lines 249 – 252: Integrate list of species into prior sentence, changing the period after “Isolates” to a colon and rewrite for clarity.

38.  Line 253: Change “from which, 19…” to “of which 19…”

39.  Line 259: Change “in addition” to “additionally”.

40.  Line 269: Change “to fight against” to “when fighting”.

41.  Line 271: Change “specially” to “particularly when”.

42.  Line 273: Insert “an” prior to “increase” and insert “in” after “increase”.

43.  Line 274: Delete the comma after “at-risk”.

44.  Line 281: Change “in the world” to “worldwide”.

45.  Line 285: Change “revision” to “review”.

46.  Line 286: Change sentence to read “…and the world by adding data…”.

47.  Line 290: Change “this review includes” to “we include”.

48.  Line 292: Insert “is” following “species”.

49.  Line 297: Change sentence to read “…our investigation revealed that one is less likely…”

50.  Lines 311 – 312: Replace wording after “very important,” to be “as outcomes of patients are often worse because of them.”

51.  Line 313: Delete “is was” and replace “revision” with “study”.

52.  Line 314: Delete the comma following “pathogens”.

53.  Line 316: Insert “and” preceding “endocarditis” and delete “among others”.

54.  Line 322: Change “there are to fight” to “when fighting”.

55.  Line 324: Change “for the” to “on”.

56.  Line 329: Delete the comma following references and insert “that are”.

57.  Line 331: Delete the comma following “treatment”.

58.  Line 333: Change “resistant” to “resistance”.

59.  Line 339: Change “revision” to “study”.

60.  Lines 342 – 343: Change sentence to read “…reported to be resistant to many current antifungals including…”.

61.  Line 349: Replace “our review shows” with “we show”.

62.  Line 350: Insert “lab” preceding “culture” and replace “than” with “that”.

63.  Line 351: Change sentence to read “culture from clinical samples remains…”.

64.  Lines 353 – 354: Change sentence to read “…identification by biochemical, proteomic, and molecular methods is generally…”

65.  Line 355: Delete commas surrounding “rather than rare”

66.  Line 361: Delete “Generally”.

67.  Line 366: Change “To finalise” to “In conclusion” and change “notice” to “note”.

68.  Line 376: Delete “A large surveillance on”.

69.  Lines 379 – 380: Change sentence to read “…laboratory personnel for better diagnosis of rare yeast infections”.

70.  Line 383: Change “or” to “and”.

71.  Lines 384 – 385: Move “is” to immediately after “Not only”.

72.  Lines 388 – 389: Change the comma to a period following “essential” and change “which” to “that”.

73.  Line 391: Change sentence to read “…pathogens has expanded, thanks to…”.

74.  Line 399: Change “addressed” to “needed”.

Only minor errors in grammar. Nicely done.

Author Response

Answer: thanks to the reviewer for the time to read and evaluate the manuscript and for noticing all the grammatical and stylistic issues. In the revised version  of the manuscript we addressed all issues mentioned by you and the other reviewers, considering that all comments were minor.

Reviewer 2 Report

Improve the bibliography in the introduction. It refers to the increase and incidence of invasive fungal diseases caused by rare yeasts has been increasing and the reference [1] is from 2011 (line 31 page 1)

Author Response

Thanks to the reviewer for the time to read and evaluate the manuscript. Following the suggestion of the reviewer, the most updated reference on infections caused by rare yeast, Chen et al 2021, was included now at the beginning of the introduction.

Reviewer 3 Report

The authors write an interesting review on rare yeasts as etiological agents describing therapy, susceptibility profile, patient characteristics (risk factors or Underlying condition) and outcome. It would be interesting to know how many articles in total they found from their initial research and how many of these did not meet the inclusion criteria.

Table 2 What do authors mean by "reference"?

Line 183. 79 cases of other infections, what are the other infections?

line 368 correct "Rhodotorula sp" in "Rhodotorula spp"

Il testo è scritto nel complesso bene, si suggerisce una revisione sui verbi.

Author Response

Comments and Suggestions for Authors

The authors write an interesting review on rare yeasts as etiological agents describing therapy, susceptibility profile, patient characteristics (risk factors or Underlying condition) and outcome. It would be interesting to know how many articles in total they found from their initial research and how many of these did not meet the inclusion criteria.

Answer: thanks to the reviewer for the time to read and evaluate the manuscript. Considering that this was a literature search rather than a systemic review and following the methodology used and reported in the last guidelines for the diagnosis and management of rare yeasts species, we did not count the total number of articles that we initially found. Once we established that an article met the inclusion criteria, we put it in a folder, and we did not consider the others.

Table 2 What do authors mean by "reference"?

Answer: In a multivariate analysis, the reference level is used as a means of interpretation of the data generated. This reference is defined by the statistical program. For instance, if “male” vs. “female” is compared, one modality should serve as reference for “Sex”.

Line 183. 79 cases of other infections, what are the other infections?

Answer: thanks to the reviewer for this question. The “other infections” were included (lines 200-201).

line 368 correct "Rhodotorula sp" in "Rhodotorula spp"

Answer: thanks to the reviewer for noticing this. However, the study by Vidotto et al. reported only 1 isolate of Rhodotorula.

Comments on the Quality of English Language

Il testo è scritto nel complesso bene, si suggerisce una revisione sui verbi.

Answer: thanks to the reviewer for the suggestion. There were few minor grammatical and stylistic issues, outlined as well by other reviewers, that were all addressed in the revised version of the manuscript.